# Improving Zero-shot
# Generalization in Offline Reinforcement Learning
# using Generalized Similarity Functions

**Bogdan Mazoure**[*]
McGill University, Quebec AI Institute
bogdan.mazoure@mail.mcgill.ca

**Ilya Kostrikov**
UC Berkeley

**Ofir Nachum**
Google Brain

**Jonathan Tompson**
Google Brain

## Abstract

Reinforcement learning (RL) agents are widely used for solving complex sequential decision making tasks, but still exhibit difficulty in generalizing to scenarios not seen during training. While prior online approaches demonstrated that using additional signals beyond the reward function can lead to better generalization capabilities in RL agents, i.e., using self-supervised learning (SSL), they struggle in the offline RL setting, i.e., learning from a static dataset. We show that performance of online algorithms for generalization in RL can be hindered in the offline setting due to poor estimation of similarity between observations. We propose a new theoretically-motivated framework called Generalized Similarity Functions (GSF), which uses contrastive learning to train an offline RL agent to aggregate observations based on the similarity of their expected future behavior, where we quantify this similarity using *generalized value functions*. We show that GSF is general enough to recover existing SSL objectives while also improving zero-shot generalization performance on two pixel-based offline RL benchmarks.

## 1 Introduction

Reinforcement learning (RL) is a powerful framework for solving complex tasks that require a sequence of decisions. The RL paradigm has allowed for major breakthroughs in various fields, e.g. outperforming humans on video games [1, 2], controlling stratospheric balloons [3] and learning reward functions from robot manipulation videos [4]. More recently, RL agents have been tested in a generalization setting, i.e. in which training involves a finite number of related tasks sampled from some distribution, with a potentially distinct sampling distribution during test time [5, 6, 7]. The main issue for designing generalizable agents is the lack of on-policy data from tasks not seen during training: it is impossible to enumerate all variants of a real-world environment during training and hence the agent must extrapolate from a (limited) training task collection onto a broader set of problems. Since the learning agent is given no training data from test-time tasks, this problem is referred to as zero-shot generalization. In our work, we are interested in the problem of zero-shot generalization where the difference between tasks is predominantly due to perceptually distinct observations. An example of this setting is any environment with distractor features [8, 9], i.e. features with no dependence on the reward signal nor the agent's decisions. This generalization setting has recently received much attention [10, 11, 12], due to its particular relevance to real-world scenarios, for example deploying the same autonomous driving agent at day or at night.

---

[*]Work done while at Google Brain.

36th Conference on Neural Information Processing Systems (NeurIPS 2022).

Generalization capabilities of an agent can be analyzed through the prism of *representation learning*, under which the agent's current belief about a rich and high-dimensional environment are summarized in a low-dimensional entity, called a representation. Recent work in online RL has shown that learning state representations with specific properties such as disentanglement [13] or linear separability [14] can improve zero-shot generalization performance. Achieving this with limited data (i.e. offline RL) is challenging, since the representation will have a large estimation error over regions of low data coverage. A common solution to mitigate this task-specific overfitting and extracting the most information out of the data consists in introducing auxiliary learning signals other than instantaneous reward [15]. As we show later in the paper, many such signals already contained in the dataset can be used to further improve generalization performance. For instance, the generalization performance of PPO on Procgen remains limited even when training on 200M frames, while generalization-oriented agents [15, 12] can outperform it by leveraging additional auxiliary signals. However, a major issue with the aforementioned methods is their exorbitant reliance on online access to the environment, an impractical restriction for real-world scenarios.

In contrast, in many real-world scenarios access to the environment is restricted to an offline, fixed dataset of experience [16, 17]. A natural limitation for generalization from offline data is that policy improvement is dependent on dataset quality. Specifically, high-dimensional problems such as control from pixels require large amounts of training experience: a standard training of PPO [18] for 25 million frames on Procgen [8] generates more than 300 Gb of data, an impractical amount of data to share for offline RL research. Improving zero-shot generalization performance from an offline dataset of high-dimensional observations is therefore a hard problem due to limitations on dataset size and quality.

In this work, we are interested in improving zero-shot generalization across a family of Partially-Observable Markov decision processes [POMDPs, 19] in an offline RL setting, i.e. by training agents on a fixed dataset. We hypothesize that in order for an RL agent to be able to generalize across perceptually different POMDPs without adaptation, observations with similar future behavior should be assigned to close representations. We use the generalized value function (GVF) framework [20] to capture future behavior with respect to any instantaneous signal (called cumulant) at a given state. Specifically, the choice of cumulant determines the nature of the behavioral similarity that is induced into state representations. For example, using reward similarity leads to learning bisimulation metrics [21, 22, 23, 24], while using future state-action visitation counts encourages reward-free behavioral similarity [25, 10, 11, 12].

Our main contributions are as follows:

1. We propose Generalized Similarity Functions (GSF), a novel self-supervised learning algorithm for reinforcement learning that aggregates latent representations of observations by their future behavior (or generalized value function).

2. Existing offline RL benchmarks [26, 27] are not well-suited to test zero-shot generalization, and so we devise two new benchmarks: offline Procgen and Distracting Control Suite. The first consists of 5M transitions from 200 related levels of 16 distinct games; the second consists of 1M transitions from 3 variations of 4 distincts tasks.

3. We evaluate performance of GSF and other baseline methods on both benchmarks, and show that GSF outperforms both previous state-of-the-art offline RL and representation learning baselines on the entire distribution of levels.

4. We analyze the theoretical properties of GSF and describe the impact of hyperparameters and cumulant functions on empirical behavior in both offline Procgen and the offline Distracting Suite benchmarks.

## 2 Related Works

**Generalization in reinforcement learning**    Generalizing a model's predictions across a variety of unseen, high-dimensional inputs has been extensively studied in the static supervised learning setting [28, 29, 30, 31]. Generalization in RL has received a lot of attention: extrapolation to unseen rewards [32, 25], observations [24, 15, 10, 11, 12] and transition dynamics [33]. Each generalization scenario is best solved by their respective set of methods: sufficient exploration [25, 34], auxiliary learning signals [35, 36, 37] or data augmentation [33, 38]. Data augmentation is a promising technique, but typically relies on handcrafted domain information, which might not be available *a priori*. In fact, we will show in our experiments that generalization in the offline RL setting is poor even when

using such handcrafted data augmentations, without additional representation learning mechanisms. In this work, we posit that representation learning should use instantaneous auxiliary signals in order to prevent overfitting onto a unique signal (e.g. reward across tasks) and improve generalization performance. Theoretical generalization guarantees have only been provided so far for limited scenarios, mostly for bandits [39], linear MDPs [40, 41, 42] and across reward functions [32, 43, 44].

**Representation learning** For simple POMDPs, near-optimal policies can be found by optimizing for the reward alone. However, more complex settings may require additional auxiliary signals in order to find state abstractions better suited for control. The problem of learning meaningful state representations (or abstractions) for planning and control has been extensively studied previously [45, 22], but saw real breakthroughs only recently, in particular due to advances in self-supervised learning (SSL). Outside of RL, SSL has achieved spectacular results by closing the gap between unsupervised and supervised learning on certain datasets [46, 47, 48, 49]. Representation learning, and specifically self-supervised learning, has also been used to achieve state-of-the-art generalization and sample efficiency results in RL on challenging control problems such as data efficient Atari [2, 50], DeepMind Control [11] and Procgen [36, 37, 15, 12]. Noteworthy instances of theoretically-motivated representation learning methods for RL include heuristic-guided learning [51], random Fourier features [42] and metric learning [52, 53].

**Offline reinforcement learning** When learning from a static dataset, agents should balance interpolation and extrapolation errors, while ensuring proper diversity of actions (i.e. prevent collapse to most frequent action in the data). Popular offline RL algorithms such as BCQ [54], MBS [55], and CQL [56] rely on a behavior regularization loss [57] as a tool to control the extrapolation error. Some methods, such as F-BRC [58] are defined only for continuous action spaces while others, such as MOReL [59] estimate a pessimistic transition model. The major issue with current offline RL algorithms such as CQL is that they are perhaps overly pessimistic for generalization purposes, i.e. CQL and MBS ensure that the policy improvement is well-supported by the batch of data.

## 3 Problem setting

### 3.1 Partially-observable Markov decision processes

A (infinite-horizon) partially-observable Markov decision process [POMDP, 19] $M$ is defined by the tuple $M = \langle \mathcal{S}, p_0, \mathcal{A}, p_{\mathcal{S}}, \mathcal{O}, p_{\mathcal{O}}, r, \gamma \rangle$, where $\mathcal{S}$ is a state space, $p_0 = \mathbb{P}[s_0]$ is the starting state distribution, $\mathcal{A}$ is an action space, $p_{\mathcal{S}} = \mathbb{P}[\cdot | s_t, a_t] : \mathcal{S} \times \mathcal{A} \to \Delta(\mathcal{S})$ is a transition function, $\mathcal{O}$ is an observation space, $p_{\mathcal{O}} = \mathbb{P}[\cdot | s_t] : \mathcal{S} \to \Delta(\mathcal{O})^2$ is an observation function, $r : \mathcal{S} \times \mathcal{A} \to [r_{\min}, r_{\max}]$ is a reward function and $\gamma \in [0, 1)$ is a discount factor. The system starts in one of the initial states $s_0 \sim p_0$ with observation $o_0 \sim p_{\mathcal{O}}(\cdot | s_0)$. At every timestep $t = 1, 2, 3, ..$, the agent, parameterized by a policy $\pi : \mathcal{O} \to \Delta(\mathcal{A})$, samples an action $a_t \sim \pi(\cdot | o_t)$. The environment transitions into a next state $s_{t+1} \sim p_{\mathcal{S}}(\cdot | s_t, a_t)$ and emits a reward $r_t = r(s_t, a_t)$ along with a next observation $o_{t+1} \sim p_{\mathcal{O}}(\cdot | s_{t+1})$.

The goal of an RL agent is to maximize the cumulative discounted rewards $\sum_{t=0}^{\infty} \gamma^t r_t$ obtained over the entire episode. Value-based off-policy RL algorithms achieve this by estimating the state-action value function under a target policy $\pi$:

$$Q^\pi(s_t, a_t) = \mathbb{E}_{\mathbb{P}_t^\pi} \Big[ \sum_{k=1}^{\infty} \gamma^k r(s_{t+k}, a_{t+k}) | s_t, a_t \Big], \tag{1}$$

for $s_t \in \mathcal{S}, a_t \in \mathcal{A}$ and where $\mathbb{P}_t^\pi$ denotes the joint distribution of $\{s_{t+k}, a_{t+k}\}_{k=1}^{\infty}$ obtained by executing $\pi$ in the environment.

An important distinction from online RL is that, in the offline RL setting, we assume access to a historical dataset $\mathcal{D}^\mu$ (instead of a simulator) collected by logging experience of the policy, $\mu$, in the form $\{o_{i,t}, a_{i,t}, r_{i,t}\}_{i=1, t=1}^{i=N, t=T}$ where, for practical purposes, the episode is truncated at $T$ timesteps. Furthermore, we assume that the agent can only be trained on a limited collection of POMDPs $\mathcal{M}_{\text{train}} = \{M_i\}_{i=1}^m$, and its performance is evaluated on the set of test POMDPs $\mathcal{M}_{\text{test}}$. We assume that both $\mathcal{M}_{\text{train}}$ and $\mathcal{M}_{\text{test}}$ were sampled from a common task distribution and that every POMDP $M_i \in \mathcal{M} = \mathcal{M}_{\text{train}} \cup \mathcal{M}_{\text{test}}$ shares the same transition dynamics and reward function with $\mathcal{M}$ but has

---

[2]$\Delta(\mathcal{X})$ denotes the entire set of distributions over the space $\mathcal{X}$.

a different observation function $p_{i,\mathcal{O}}$. Importantly, since we perform control from pixels, we are in the POMDP setting [see 60] and therefore emphasize the difference between observations $o_t$ and corresponding states $s_t$ throughout the paper.

## 3.2 Representation learning

Previous works in the RL literature have studied the use of auxiliary signals to improve generalization performance. Among others, [10, 11] define the similarity of two observations to depend on the distance between action sequences rolled out from that observation under their respective optimal policies. They achieve this by finding a latent space $\mathcal{Z} \subseteq \mathcal{S}$ in which the distance $d_{\mathcal{Z}}(z, z')$ for all $z, z' \in \mathcal{Z}$ is equivalent to distance between true latent states $d_{\mathcal{S}}(s, s')$ for all $s, s' \in \mathcal{S}$; the aforementionned works learn $\mathcal{Z}$ by optimizing action-based similarities between observations. In practice, latent space $z$ is decoded from observation $o$ using a latent state decoder $f : \mathcal{O} \to \mathcal{Z}$ from observation $o_t$. Throughout the paper, we assume that all value functions have a linear form in the latent decoded state, i.e. $Q_\theta(o,a) = \theta_a^\top f_\psi(o) = \theta_a^\top z_\psi$, which agrees with our practical implementation of all algorithms. Within this model family, the ability of an RL agent to correctly decode latent states from unseen observations directly affects its policy, and therefore, its generalization capabilities. In the next section, we discuss why representation learning is important for offline RL, and how existing action-based similarity metrics fail to recover the true latent states for important families of POMDPs.

# 4   Motivating example

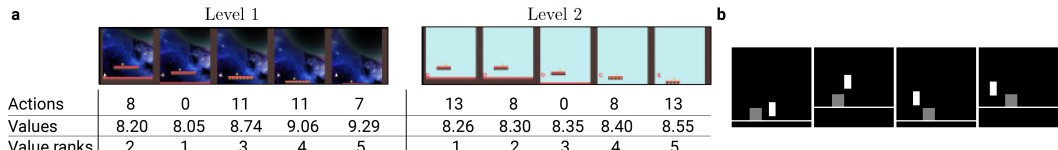

| | Level 1 | | | | | Level 2 | | | | |
|---|---|---|---|---|---|---|---|---|---|---|
| Actions | 8 | 0 | 11 | 11 | 7 | 13 | 8 | 0 | 8 | 13 |
| Values | 8.20 | 8.05 | 8.74 | 9.06 | 9.29 | 8.26 | 8.30 | 8.35 | 8.40 | 8.55 |
| Value ranks | 2 | 1 | 3 | 4 | 5 | 1 | 2 | 3 | 4 | 5 |

Figure 1: **(a)** Two levels of the Climber game from the Procgen benchmark [8] with *near-identical* true latent states and *near-identical* value functions but *drastically different* action sequences. **(b)** Four levels of the jumping task [61] where the constant reward signal makes policy similarity more informative than state value similarity.

Multiple recently proposed self-supervised objectives [10, 11] conjecture that observations $o_1 \in M_1, o_2 \in M_2$ that emit similar future action sequences under optimal policies $\pi_1^*, \pi_2^*$ should be decoded into nearby latent states $z_1, z_2$. While this heuristic can correctly group observations with respect to their true latent state in simple action spaces, it fails to identify similar pairs of trajectories in POMDPs with multiple optimal policies[3]. For instance, two trajectories might visit an identical set of latent states, but have drastically different actions.

Fig. 1a shows one such example: two levels of the Climber game have a near-identical true latent state (see Appendix) and value function (average normalized mean-squared error of 0.0398 across episode), while having very different action sequences from a same PPO policy (average total variation distance of 0.4423 across episode). The problem is especially acute in Procgen, since the PPO policy is high-entropy for some environments (see Fig. 5), i.e. various levels can have multiple drastically different near-optimal policies, and hence fail to properly capture observation similarities.

In this scenario, assigning observations to a similar latent state by value function similarity would yield a better state representation than reasoning about action similarities. On the other hand, Fig. 1b shows a domain where grouping state representations by action sequences can be optimal. So how do we unify these similarity metrics under a single framework? In the next section, we use this insight to design a general way of improve representation learning through self-supervised learning of discounted future behavior.

---

[3]While optimal policies are not guaranteed to be unique, the optimal value function is unique.

## 5 Method

We propose measuring a generalized notion of future behavior similarity using generalized value functions, as defined by the corresponding cumulant function. The choice of cumulant determines which components of the future trajectory are most relevant for generalization.

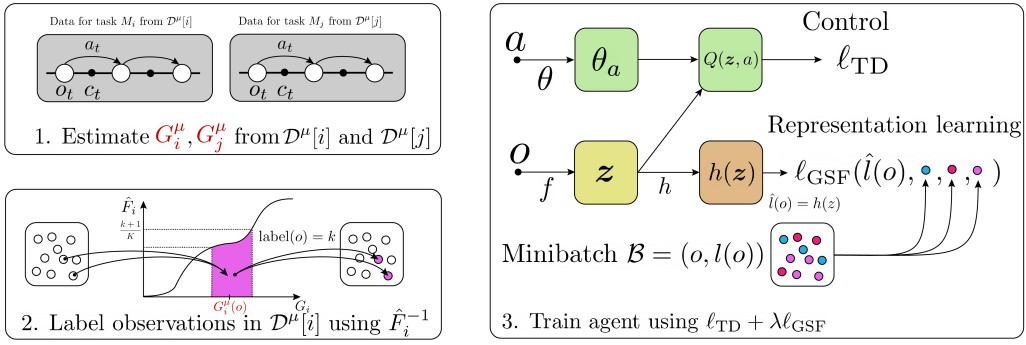

Figure 2: Schematic view of GSF : the offline dataset $\mathcal{D}^\mu$ is used to estimate POMDP-specific GVFs wrt some cumulant function $c$, whose quantiles are then used to label each observation in the dataset.

### 5.1 Quantifying future behavior with GVFs

An RL agent's future discounted behavior can be quantified not only by its value function, but other auxiliary signals, for example, by its future observation occupancy measure, known as successor representation [62, 32]. The choice of the signal used during value iteration measures the properties the agent will exhibit in the future, such as accumulated returns, actions, or observation visitation density. See Thm. 1 in the Appendix for the connection between successor features and interpolation error in our method.

Following the work of [20], we can broaden the class of value functions to any kind of cumulative discounted signal, as defined by a bounded cumulant function $c : \mathcal{O} \times \mathcal{A} \to \mathbb{R}^d$, s.t. $|c(o,a)| \leq c_{\max}$ for $c_{\max} = \sup_{o,a \in \mathcal{O} \times \mathcal{A}} c(o,a)$. While typically cumulants are scalar-valued functions (e.g. reward), we also make use of the vector-valued case for learning the successor features [32], in which case the norm of $c(o,a)$ is bounded.

**Definition 1 (Generalized value function)** *Let $c$ be any bounded function over $\mathbb{R}^d$, let $\gamma \in [0,1]$ and $\mu$ any policy. The generalized value function is defined, for any timestep $t \geq 1$ and $o_t \in \mathcal{O}$, as*

$$G^\mu(o_t) = \mathbb{E}_{\mathbb{P}_t^\mu}\Big[\sum_{k=1}^\infty \gamma^k c(o_{t+k}, a_{t+k}) | o_t\Big]. \tag{2}$$

Since, in our case, we can learn $G^\mu$ for each distinct POMDP $M_i$ for the dataset $\mathcal{D}^\mu$, we index the GVF using the POMDP index, i.e. $G_i^\mu = \texttt{LearnGVF}(c, \mathcal{D}^\mu, i)$ (in practice, learning is parallelized).

---

**Algorithm 1:** $\texttt{LearnGVF}(c, \mathcal{D}^\mu, i, \theta^{(0)}, J, \alpha, \gamma)$: Offline estimation of GVF $\hat{G}_i^\mu$

**Input** : Cumulant function $c$, dataset $\mathcal{D}^\mu$, POMDP label $i$, initial parameters $\theta^{(0)}$, target parameters $\tilde{\theta}$, latent state decoder $f$, iterations $J$, learning rate $\alpha$, discount $\gamma$

1 **for** $j = 1, .., J$ **do**
2     $o, a, o' \sim \mathcal{D}[i]$; // Sample transition from subset corresponding to POMDP $i$
3     $c \leftarrow c(o, a)$;
4     $o \leftarrow \texttt{random crop}(o)$;
5     $z, z' \leftarrow f(o), f(o')$;
6     $\theta^{(j)} \leftarrow \theta^{(j-1)} - \alpha \nabla_{\theta^{(j-1)}} (G_{\theta^{(j-1)}}(z) - c - \gamma G_{\tilde{\theta}^{(j-1)}}(z'))^2$ ;
7     Update target parameters $\tilde{\theta}$ with $\beta$ of online parameters $\theta$;

---

## 5.2 Measuring distances between GVFs of different POMDPs

Examining the difference between future behaviors of two observations quantifies the exact amount of expected behavior change between these two observations. Using the GVF framework, we could compute the distance between $o_1 \in M_1$ and $o_2 \in M_2$ by first estimating the latent state with $z = f(o)$ using a latent state decoder $f$, and then using the following distance as a measure of dissimilarity

$$d_\mu(o_1{}^i, o_2{}^j) = ||G_i^\mu(f(o_1)) - G_j^\mu(f(o_2))||, i, j = 1,... \tag{3}$$

However, the distance between GVFs from two different POMDPs can have drastically different scales, i.e. $\sup_{o_1, o_2} |G_1^\mu(o_1) - G_2^\mu(o_2)| \leq \frac{c_{1,\max}^\mu + c_{2,\max}^\mu}{1 - \gamma}$, thus making point-wise comparison meaningless. The issue is less acute for cumulants which are homogenous between different POMDPs (e.g. indicator functions for successor representation), and more problematic when the cumulant incorporates a more heterogeneous signal, such as the extrinsic reward function. To avoid this problem, we suggest performing a comparison based on order statistics.

Namely, a robust distance estimate between GVF signals across POMDPs can be obtained by looking at the cumulative distribution function of $G_i$ denoted $F_i(g) = \mathbb{P}[G_i(o_t) \leq g]$ for all $o_t \in \mathcal{O}$. $G_i$ is a deterministic GVF with the set of discontinuity points of measure 0, and as such $F_i$ can be understood through the induced state distribution $\mathbb{P}_t^\mu$ (using continuous mapping theorem from [63]). It can be estimated from $n$ independent and identically distributed samples of $\mathcal{D}^\mu$ as

$$\hat{F}_i(g) = \frac{1}{n} \sum_{i=1}^n \mathbb{1}_{G_i < g}, G_i = \texttt{LearnGVF}(c, \mathcal{D}^\mu, i), g \in \left[-\frac{c_{i,\max}}{1-\gamma}, \frac{c_{i,\max}}{1-\gamma}\right] \tag{4}$$

and its inverse, the empirical quantile function [64]

$$\hat{F}_i^{-1}(p) = \inf\{g \in \left[-\frac{c_{i,\max}^\mu}{1-\gamma}, \frac{c_{i,\max}^\mu}{1-\gamma}\right] : p \leq F_i(g)\}, \tag{5}$$

for $p \in [0,1]$. We use the empirical quantile function to partition the range of all GVFs into $K$ quantile bins, i.e. disjoint sets with identical size where the set corresponding to quantile $k$ is defined as $I_i(k) = \{o \in M_i : F_i^{-1}(\frac{k}{K}) \leq G_i^\mu(o) \leq F_i^{-1}(\frac{k+1}{K})\}$ and its aggregated version as $I(k) = \cup_{i=1}^m I_i(k)$[4]. Importantly, we augment the dataset $\mathcal{D}^\mu$ with observation-specific labels, which correspond to the index of the quantile bin into which the GVF $G$ of an observation $o \in M_i$ falls into $l_i(o) = \max_k \mathbb{1}_{o \in I_i(k)}$.

These self-supervised labels are then used in a multiclass InfoNCE loss [47], which is a variation of metric learning with respect to the quantile distance defined above [65, 66] and this forms the basis of our self-supervised learning objective.

## 5.3 Self-supervised learning of GSFs

After augmenting the offline dataset with observation labels as described above, we use a simple self-supervised learning procedure to minimize distance in the latent representation space between observations with identical labels.

First, the observation $o$ is encoded using a non-linear encoder $f_\psi : \mathcal{O} \to \mathcal{Z}$ with parameters $\psi$ into a latent state representation $z = f_\psi(o)$[5]. The representation $z$ is then passed into two separate trunks: 1) a linear matrix $\theta_a$ which recovers the state-action value function $Q_\theta(o,a) = \theta_a^\top z$, and 2) a non-linear projection network $h_\theta : \mathcal{Z} \to \mathcal{Z}$ with parameters $\theta_h$ to obtain a new embedding, used for self-supervised learning. The projection $h_\theta(z)$ is then used in a multiclass InfoNCE loss [47, 66] where a linear classifier $\mathbf{W} \in \mathbb{R}^{|\mathcal{Z}| \times K}$ aims to correctly predict the observation labels (i.e. quantile bins $k = 1,2,..,K$) from $h_\theta(z)$, for temperature $\tau > 0$:

$$\ell_{\text{GSF}}(\theta, \psi, \mathbf{W}) = -\mathbb{E}_{o \sim \mathcal{D}^\mu} \left[ \sum_{k=1}^K \mathbb{1}_{l(o)=k} \text{LogSoftmax}(\mathbf{W}^\top h_\theta(f_\psi(o))/\tau)_k \right]. \tag{6}$$

---

[4]A special case of quantile binning occurs when $K = n$, in which case the auxiliary task is to predict the rank of the GVF associated to a given observation in the current minibatch.

[5]This encoder is different from the one used to evaluate the GVFs.

## 5.4 Full Algorithm

Given $m$ training tasks, GSF first learns GVF estimates $G_1^\mu, .., G_2^\mu$ by applying `LearnGVF` to task-specific data from the offline dataset $\mathcal{D}^\mu$. Each data point in $\mathcal{D}^\mu$ is then labeled with the quantile into which its GVF falls. These labels are then used to jointly optimize Eq.6 and a control loss with respect to the encoder parameters. To learn the value function $Q_\theta$, we use CQL [56], which is trained using a linear combination of Q-learning [67, 16] and behavior-regularization:

$$\ell_{\text{CQL}}(\theta) = \mathbb{E}_{o,a,r,o'\sim\mathcal{D}^\mu}[(r+\gamma\max_{a'\in\mathcal{A}}Q_{\tilde{\theta}}(o',a')-Q_\theta(o,a))^2] + \lambda\mathbb{E}_{s\sim\mathcal{D}^\mu}[LSE(Q_\theta(o,a))-\mathbb{E}_{a\sim\mu}[Q_\theta(o,a)]], \quad (7)$$

for $\lambda \geq 0$, $\tilde{\theta}$ target network parameters[6] and $LSE$ being the log-sum-exp operator [7]. For domains with continuous actions, we also decode the Boltzmann policy $\pi$ from $Q_\theta$.

Fig. 2 provides a schematic view of the algorithm, while Alg. 2 in the Appendix presents the exact learning procedure for GSF as implemented on top of a CQL agent for a discrete action space.

**Recovering existing self-supervised objectives** The generality of our framework allows it to recover existing objectives such as CSSC and PSEs by carefully designing the cumulant function. Below, we highlight which existing algorithms can be recovered by GSF.

- **Cross-State Self-Constraint [CSSC, 10]**: In CSSC, observations $o_1, o_2$ are considered similar if they have identical future action sequences of length $K$ under some fixed policy; a total of $|\mathcal{A}|^K$ distinct classes are possible. This approach can be approximated in our framework by picking $c(o_t, a_t) = \mathbb{1}_{a_t}(a), \forall a \in \mathcal{A}$. The problem reduces to a $|\mathcal{A}|^{T-t}$-way classification problem for observations of timestep $t$, which GSF approximates using $K$ quantiles.

- **Policy similarity embedding [PSE, 11]**: PSEs balance the distance between local optimal behaviors and long-term dependencies in the transitions, notably using $d_{\text{TV}}$. If we consider the space of Boltzmann policies $\pi_{\text{Boltzmann}}$ with respect to a POMDP-specific value function $Q$, then choosing $c(o_t, a_t) = r(s_t, a_t)$ in GSF will effectively compute the distance between unnormalized policies.

**The choice of $K$ induces a bias-variance trade-off** How should the number of quantiles $K$ (read labels) be set, and what is the effect of smaller/ larger values of $K$ on the learned representations? Thm. 1 highlights a trade-off when choosing the number of quantile bins empirically.

**Theorem 1** *Let $G_1$, $G_2$ be generalized value functions with cumulants $c_1, c_2$ from respective POMDPs $M_1, M_2$, $K$ be the number of quantile bins, $n_1, n_2$ the number of sample transitions from each POMDP. Suppose that $\mathbb{P}[\sup_{t=1,2,..}|c_1(o_{1,t}, \mu(o_{1,t})) - c_2(o_{2,t}, \mu(o_{2,t}))| > (1-\gamma)\varepsilon/\gamma] \leq \delta$. Then, for any $k=1,2,..,K$ and $\varepsilon > 0$ the following holds without loss of generality:*

$$\mathbb{P}\left[\sup_{o_1,o_2\in I(k)}|G_1(o_1)-G_2(o_2)| > 3\varepsilon\right] \leq 2e^{-2n_1\varepsilon^2/4} + \mathbb{P}\left[\sup_{k=1,2,..,K}\left|\hat{F}_1^{-1}\left(k+1/K\right)-\hat{F}_1^{-1}\left(k/K\right)\right| > \varepsilon\right] + p(n_1,K,\varepsilon) + \delta \quad (8)$$

The proof can be found in the Appendix Sec. A.5. For POMDP $M_1$, the error decreases monotonically with increasing bin number $K$ (second term) but the variance of bin labels depends on the number of sample transitions $n_1$ (first term). The inter-POMDP error (third term) does not affect the bin assignment. Hence, choosing a large $K$ will amount to pairing states by rankings, but results in high variance, as orderings are estimated from data and each bin will have $n=1$. Setting $K$ too small will group together unrelated observations, inducing high bias.

**Limitations** As is the case with all offline RL methods, GSF is limited by the compounding extrapolation error under low data coverage. We hypothesize that wise choices of $K$ and $c$ can mitigate the extrapolation error by learning observation groups with low intra-group variance, but, since they are environment-dependent, searching for an optimal $(K,c)$ pair can be computationally expensive.

---

[6]A copy of $\theta$ updated solely using an exponential moving average (see Appendix).
[7]https://en.wikipedia.org/wiki/LogSumExp

## 6 Experiments

Unlike for single task offline RL [26], most prior work on zero-shot generalization from offline data either come up with an *ad hoc* solution suiting their needs, e.g. [33], or assess performance on benchmarks that do not evaluate generalization across observation functions [e.g., 68]. To accelerate progress in this field, we devised two benchmarks: offline Procgen (discrete actions) and offline Distracting Suite (continuous actions) - two offline RL datasets to directly test for generalization of RL agents across observation functions[8]. Moreover, for a standard comparison, we provide generalization results on the classical online Procgen simulator, comparing PPO to PPO with GSF and PPO with PSE, respectively.

**GVF training**    In the offline setting, the training dataset is used to learn a set of task-specific GVFs in parallel as follows. First, we project each observation $o$ into a latent representation $z = f_{\psi'}(o)$; we then pass $z$ through a non-linear network $h_{\theta'} : \mathcal{Z} \rightarrow \mathbb{R}^{d_c \times m}$ where $d_c$ is the dimensionality of the cumulant function's output. The output of $h_{\theta'}$ is then split into $m$ disjoint chunks, which are in turn used in their respective temporal difference losses $\ell_{\text{TD}}$ in place of value functions. The procedure can be adapted to an online setting via a similar procedure, except that all GVF estimators are trained simultaneously and independently in separate simulators.

**Offline Procgen benchmark**    We evaluate the proposed approach on an offline version of the Procgen benchmark [8], which is widely used to evaluate zero-shot generalization across complex visual perturbations. Given a random seed, Procgen supports sampling procedurally generated level configurations for 16 games under various complexity modes: "easy", "hard" and "exploration". More details can be found in Appendix.

**Offline Procgen results**    We compare the zero-shot performance on the entire distribution of "easy" POMDPs for GSF against that of strong RL and representation learning baselines: behavioral cloning (BC) - to assess the quality of the PPO policy, CQL [56] - the current state-of-the-art on multiple offline benchmarks which balances RL and BC objectives, CURL [35], CTRL [12], DeepMDP [69] - which learns a metric closely related to bisimulation across the MDP, Value Prediction Network [VPN, 70] - which combines model-free and model-based learning of values, observations, next observations, rewards and discounts, Cross-State Self-Constraint [CSSC, 10] - which boosts similarity of observations with identical action sequences, as well as Policy Similarity Embeddings [34], which groups observation representations based on distance in optimal policy space.

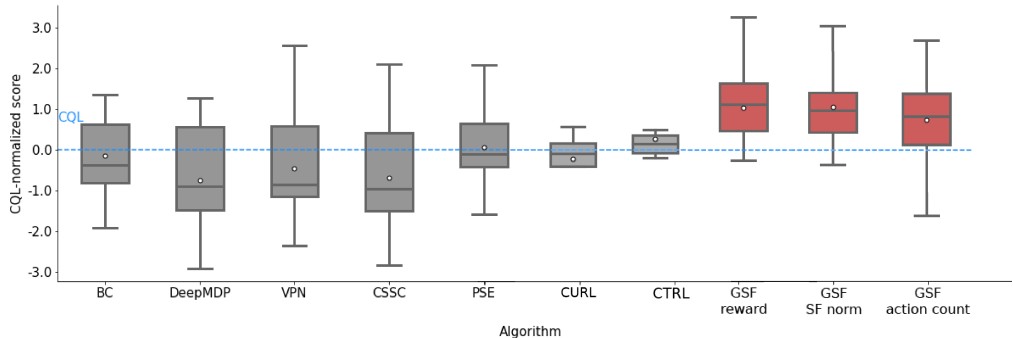

Figure 3: Returns on the offline Procgen benchmark [8] after 1M training steps. Boxplots are constructed over 5 random seeds and all 16 games; each method is normalized by the per-game median CQL performance. White dots represent average of distribution.

Fig. 3 shows the performance of all methods over 5 random seeds and all 16 games on the offline Procgen benchmark after 1 million gradient steps. Per-game average scores for all methods can be found in Tab. 2 (Appendix). The scores are standardized per-game using the downstream task's (offline RL) performance, in this case implemented by CQL. It can be seen that GSF performs better than other offline RL and representation learning baselines.

---

[8]Code can be found at https://github.com/bmazoure/gsf_public.

Using different cumulant functions can lead to different label assignments and hence different similarity groups. Fig. 3 examines the performance of GSF with respect to 3 cumulants: 1) $r(s_t, a_t)$, rewards s.t. GSF learns the policy's $Q^\mu$-value, 2) $\mathbb{1}_{o_t}(o)$, the successor representation[9] [62, 32] s.t. GSF learns the distribution induced by $\mu$ over $\mathcal{D}^\mu$ [71] and 3) $\mathbb{1}_{a_t}(a)$, action counts, s.t. GSF learns discounted policy. While rewards and successor feature cumulant choices leads to similar performance, using action-based distance leads to larger variance.

**Offline Distracting Control Suite results** Following the same procedure as for offline Procgen results, we first formed an offline dataset from the challenging Distracting Control Suite [9] by saving the replay buffer of Soft Actor-Critic [72] trained for 1M frames on 4 environments with 2 different background perturbations. Next, we pre-trained $G_1^\mu, G_2^\mu$ with action and reward-based cumulants, which were then used in conjunction with CQL to learn a single multi-task policy. Fig 4 shows the online performance of GSF evaluated on 10 background perturbations not seen during training, normalized by per-environment median CQL scores.

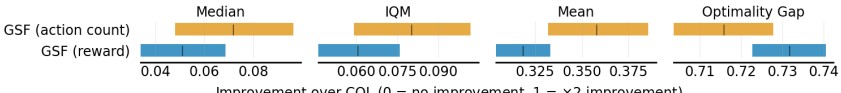

Figure 4: Improvement of GSF over CQL with action and reward-based similarity functions aggregated using performance metrics on 3 seeds and 4 environments of the Distracting Control Suite reported as suggested by [73] with 95% confidence intervals. We can see that using action counts results in higher mean, median and interquartile mean (IQM) statistics and lower optimality gap (i.e. fraction of scores falling under a certain threshold), than when using rewards, or when comparing with the performance of CQL.

The results are consistent with findings of [11], in that 1) learning policy-based similarity improves generalization capabilities of state representations, and 2) unlike in Procgen, policy-based similarity provides a better learning signal than value-based similarity.

**Online Procgen results** We additionally compare performance of GSF to that of PPO and PPO with PSEs in the classical online Procgen benchmark [5]. Figure 8 shows test returns for 20M frames obtained on the entire distribution of easy levels while training on 200 easy levels for PPO, PPO+PSEs [11] and our PPO+GSFs. PPO+GSF outperforms or matches both PPO and PPO+PSE on most environments, showing that GSF can be efficiently combined with both offline and online algorithms. See Appendix A.6.1 for detailed results.

## 7 Discussion

In this work we proposed Generalized Similarity Functions, a novel framework which combines reinforcement learning with representation learning to improve zero-shot generalization performance on challenging, pixel-based control tasks. GSF relies on computing the similarity between observation pairs with respect to any instantaneous accumulated signal, which leads to improved empirical performance on two newly introduced benchmarks, offline Procgen and offline Distracting Suite. Theoretical results suggest that GSF's hyperparameter choice depends on a bias-variance trade-off.

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
