# OpenReview forum: "Improving Zero-Shot Generalization in Offline Reinforcement Learning using Generalized Similarity Functions"
_NeurIPS.cc/2022/Conference — NeurIPS 2022 Accept_

### Official Review · Reviewer_dP54 · 2022-07-10

**Rating:** 6
**Confidence:** 2
**Ethics Flag:** Yes
**Soundness:** 3 good
**Presentation:** 2 fair
**Contribution:** 3 good

**Summary:**

The paper proposes a novel representation learning method Generalized Similarity Functions (GSF) to improve the generalization ability of offline RL. GSF clusters latent states with similar future behaviors quantified by the order statistics of Generalized Value Function (GVF). Empirical results on the offline Procgen benchmark and the offline Distracting Control Suite show that GSF successfully enhances the performance on unseen tasks.

**Questions:**

* Q1: The authors may need to spend more effort on the related works such as [1][2].
* Q2: The authors are suggested to add a TSNE plot of the latent states of different training and testing tasks.
* Q3: A comparison of the computation cost is needed.


**Typos:**
1. line 185: you may need to delete "the" in "by its the value function".
2. line 233: two repeated words "used".


**Ethics Review Area:**

["I don’t know"]

**Limitations:**

The authors state that the limitation of the proposed method is the search of hyper-parameters.

**Strengths And Weaknesses:**

**Strengths:**
* The problem of zero-shot generalization in offline RL is significant, and the idea of GSF seems novel.
* The choice of hyper-parameters has some theoretical analysis.
* The paper conducted experiments in both discrete and continuous benchmarks, and the experimental results in both offline and online settings demonstrate the effectiveness of the proposed method.

**Weaknesses:**
* Some related works are missing. [1] and [2] both introduce representation clustering methods for offline multi-task/meta RL and can generalize to unseen tasks.
* There is no visualization of the learned representations of different training and testing tasks, which could help readers better understand the effectiveness of GSF.
* The paper does not discuss the computation cost of GSF.

[1]  Li J, Vuong Q, Liu S, et al. Multi-task batch reinforcement learning with metric learning[J]. NeurIPS 2020.
[2] Li L, Yang R, Luo D. Focal: Efficient fully-offline meta-reinforcement learning via distance metric learning and behavior regularization. ICLR 2021.

---

> ### Author Response · Authors · 2022-08-01
> **Response to reviewer dP54**
>
> Thank you for your insightful comments, and we apologize for omitting these previous works. We provide detailed point-by-point answers to your questions below.
>
> * **Missing related works:** We apologize for overlooking these previous papers - we include them in the paper's current revision.
>
> * **Visualization of tasks:** In our case, GSFs do provide a partial ordering on all observations in the dataset via the quantiles of the GVF function. Following your suggestion, we have conducted a qualitative analysis of the state representations learned by PPO vs PPO+GSF algorithms, which can be found in Appendix A.6.2 and shows how adding auxiliary predictive losses can impose a stronger structure on the state representations learned by the PPO agent.
>
> * **Computational cost of GSF:** The time complexity of learning representations using GSFs can be split into two phases: pre-training of GVF functions, which has the same complexity as off-policy evaluation with function approximation (see *Average-reward off-policy policy evaluation with function approximation* by Zhang et al. (2021)), and computing self-supervised task labels, which involves finding the empirical quantile function. Constructing the EQF involves ranking all data points in the dataset according to their GVF, which can be done in log-linear time. Finally, the bulk of the computational cost is taken by updating the agent’s parameters with respect to the control and self-supervised loss, which involves computation of large gradient tensors but is also done by all baseline algorithms.
>
> * **Typos:** Thank you for noting them, we have corrected them in our revision.

---

> > ### Comment · Reviewer_dP54 · 2022-08-04
> > **Additional questions about the new visualization results**
> >
> > Thank you for your response! The new visualization result is interesting. I have some additional questions about the new result.
> >
> > * Since the major contribution of the paper is zero-shot generalization in offline RL, why would you use an agent trained online for visualization? How about the results of agents trained offline?
> >
> > * Are the generated 100 trajectories encountered during pretraining or are they zero-shot generalization tasks?
> >
> > * Can you also visualize the corresponding observations (similar to the Figure 1 in Section 4) of some grouped points in the UMAP plots?
> >
> > * I think you can further discuss how such representation (in Figure 10) helps with zero-shot generalization.

---

> > > ### Author Response · Authors · 2022-08-05
> > > **Clarification regarding visualization experiments**
> > >
> > > Thank you for the fast response - we appreciate your concerns and provide detailed answers below.
> > >
> > > * **Online vs offline UMAP results:** We indeed first provided UMAP results in the online setting, which was due to a technical aspect that made it easier to run online experiments during the rebuttal phase for us. We have now added the results for UMAP in the offline setting for GSF as Figure 11 and 12 in the Appendix of the revised manuscript. You can see that a pattern similar to the online setting holds, although it is perhaps less smooth, which we hypothesize is due to the out-of-distribution action selection problem in offline RL.
> > >
> > > * **Source of visualized data points:** Apologies for not clarifying this directly in the revision: all UMAP figures are obtained by first training the agents on samples from the set of training tasks of the Plunder game, and then collecting 100 trajectories on the test set of tasks. So both figures show trajectories from the test set of POMDPs.
> > >
> > > * **Visualization of sample observations:** This is a good suggestion, since it provides an intuitive mechanism to inspect the nature of the state abstractions learned by both RL agents. We have updated both figures in the latest revision to show observations along two test trajectories. Specifically, we overlaid three sample observations from two sample trajectories obtained on the test set of tasks, and show their approximate location in the space found by UMAP. We can immediately see that GSF representations are ordered not only according to their value function, but also temporally: these observations are mapped close to each other pairwise (e.g. higher value states are mapped close to each other for all tasks). This implies that the policy found by the GSF agent would behave similarly independently of the task-specific information, which implies better robustness to distracting features such as changing background images, color swaps, etc.
> > >
> > > * **Usefulness of the GSF representation as per UMAP results:** The core insight that our work provides via the lens of GSF is that zero-shot generalization capabilities of RL agents can be improved by structuring the space of latent state representations according to some similarity metrics. This concept of mapping “similar” observations to points which are close in this space is fundamental to other areas of machine learning and deep learning, e.g. self-supervised algorithms such as SwAV (Caron et al. 2020) or MYOW (Azabou et al. 2021) leverage clustering or *k*-nearest neighbors information in order to learn such metric spaces; as their results have shown, this idea is extremely powerful and can reduce the gap between performance of supervised and unsupervised learning methods on some tasks. In reinforcement learning, similarity metrics which are used to aggregate state representations often have to be time-dependent, e.g. be it via action sequences or reward sequences. Looking at the updated Figure 10, observations from different POMDPs are grouped by their value function and not their task id, meaning that training with this similarity metric will effectively teach the encoder to focus on task-agnostic features, e.g. ignore the color of the agent. We've added some of this discussion in the Appendix section on UMAP results of the latest revision.

---

### Official Review · Reviewer_EVg5 · 2022-07-10

**Rating:** 6
**Confidence:** 3
**Soundness:** 3 good
**Presentation:** 2 fair
**Contribution:** 3 good

**Summary:**

This paper proposes a framework called Generalized Similarity Functions (GSF), which estimates the similarity between observations by the various future behaviors. The previous works tried to calculate the similarity based on the future action sequences, state distribution, value, etc; but the adequateness of those methods depends on the problem, and no method is optimal in all circumstances. To this end, GSF learns from some arbitrary c(s,a) function and results in a value functions of those functions. Authors argue that the proposed framework can include the previous methods by using appropriate c(s,a). After learning GSF, the paper augments the data by additional labels, i.e. quantile of GSF, and learns the latent space by minimizing InfoNCE loss.

**Questions:**

- It seems $c(s,a)$ is proposed to be a vector-valued function to recover some cases, e.g. successor features, CSSC. $G$ is a cumulative sum of $c$, so it will be vector-valued when $c$ is vector-valued. However, below definition 1, it is explained that $G_i$ is learned for each distinct POMDP. Is it only the case when $G$ is learned based on reward? Or is $G$ matrix-valued when $c$ is vector-valued? Either way, the description does not seem very clear.

- The aggregated version $I(k)=\cup_{i=1}^m I_i(k)$ does not seem to be used. The definition of $l_i(o)=\max_k \mathbb{1}_{o\in I_i(k)}$ is also strange; is it $\arg \max$? Based on a number of quantiles based on different dimension of $G$, how exactly is $l_i$ computed?

- It is hard to understand how CSSC is recovered (mainly due to the lack of understanding on $l_i$). Can it be exactly recovered when we have large enough $K$? Can you also elaborate more on how to recover PSE precisely?

- In the experiment section, the proposed framework shows much performance improvement on the Procgen benchmark (almost double the performance of CQL), whereas other algorithms perform much worse than CQL, or are barely comparable. Is GSF the only algorithm that is based on CQL? What happens if we use other representation learning algorithms with CQL? (or is it already done in that way?)

- On the other hand, in distracting control suite, GSF only shows a very small improvement (~=4%). What is the main reason for being this bad compared to the Procgen benchmark?

- What happens if we use all these different similarity metrics at the same time? Can the algorithm find out the best one to use? Can we know what similarity metric to use in advance?

**Limitations:**

The authors adequately addressed the limitations and potential negative social impact of their work.

**Strengths And Weaknesses:**

- This paper proposes a framework that can capture any signal and use it to train the representation. It is a generalized framework that can recover a number of previous frameworks. In that sense, it is somewhat original but not something that is completely new.

- The paper seems to have OK-ish technical quality, but some stuff is not clearly written and is hard to follow (e.g. definitions of $c$, $G$, $F$). The experiment session is also somewhat not easy to understand (what is the optimality gap in figure 4?)

- This paper will have some impact on the researchers in this field, due to the good performance and the proposal of the benchmarks.

---

> ### Author Response · Authors · 2022-08-01
> **Response to reviewer EVg5**
>
> Thank you for your comments - we address your concerns in our response below. We will add clarifications on the technical aspects of the paper in the final version, and if you have specific questions or concerns regarding specific technical parts, please let us know.
>
> * **Regarding the GVF structure:** You are correct to assume that since cumulants can be scalar or vector-valued functions so are the GVFs. The best example of a vector-value GVF is the successor representation, where the one-step TD update is essentially performed across all dimensions of the state representation. The problem, as you pointed out, arises when needing to compute the similarity, or rather, the quantile function. We propose one solution to this that introduces approximations: for example, we can approximate count-based methods by noting that $\frac{\gamma}{n(o)+1}-\frac{\gamma^2}{1-\gamma}\leq (1+\gamma)-||G(o)||_1\leq\frac{\gamma}{n(o)+1}$ (see Machado et al. 2020), and hence it is possible to reduce the vector-valued $G(o)$ to a scalar approximation to count-based methods. A similar reasoning can be applied for CSSC, where we can keep track of action counts via a one-hot vector (for discrete action spaces).
>
> * **Aggregate bin definition:** In practice, we use $I_i(k)$ when assigning quantile labels to all GVF signals for a task $i$. In theory, we use $I(k)$ to denote all observations across all tasks whose GVFs fall into the $k^\text{th}$ quantile, which is useful to show that the spread between two tasks’ GVF quantiles can be bounded based on the cumulant choice. Moreover, $I(k)$ is used in practice during training, as when sampling a batch of data, we have to assign label k to an observation $o\in I(k)$.
>
> * **Connection to CSSC and PSE:** PSE and CSSC both measure similarity between observations in the action space. That is, given two policies, we either measure their divergence in the current and future timesteps analytically or empirically. If we choose a sample-based approximation with $n=1$, we can sample $a_1 \sim \pi_1(o),a_2\sim \pi_2(o)$ and compute some divergence based on that (high variance). PSEs have a recursive update which also incorporates future timesteps into this equation. Unrolling it yields a quantity proportional to the sum of future single-step policy divergences averaged across i.i.d. rollouts. This is similar to what GSFs learn under action-based similarity by doing one-step TD updates on one-hot encoded action counts wrt $\pi_1,\pi_2$ and then binning the expected future action counts by quantiles.
>
> * **CQL experiments:** All of the methods follow the exact same experimental backbone, i.e. use CQL with data augmentation of random crops + auxiliary loss. We carried out the following protocol for generating the boxplots: we aggregated performance across all 16 Procgen games and all random seeds and standardized all scores by per-game averages of CQL. So the higher end of the boxplots in Fig.3 tend to be a fraction of the games where GSFs help a lot, while the lower end of boxplots are the ones where it has no effect.
>
> * **Distracting Control Suite:** The reason why the improvement is so different between Procgen and DCS is the following: the structure of the Distracting Control Suite is such that CQL already performs close to optimal on the 4 selected tasks since SAC was trained for ~1M timesteps and performance is saturated on this benchmark. Note that our results are consistent with Agarwal et al. (2021) Table 3, where the improvement is also small on the Distracting Control Suite and large on other domains.
>
> * **Combining multiple similarities:** Due to the flexibility of GSFs, it is indeed possible to combine various forms of signals, e.g. reward and action based similarities. The main issue, as usual in such cases, comes from the relative scaling of two signals with respect to one another. For example, for deterministic policies on DCS the variance of the corresponding GVF signal across states might be much smaller than that of the value function. To avoid this, it is possible either to re-normalize all GVFs within some bounded range (e.g. $[\frac{c_\text{min}}{1-\gamma},\frac{c_\text{max}}{1-\gamma}]$, or introduce a trade-off hyperparameter s.t. $c_\text{composite}(o)=c_\text{action}(o)+\lambda_\text{composite} c_\text{reward}(o)$. We have not studied this approach empirically, but we think it might be interesting to try out.
>
> * **Choice of cumulant:** It is possible to find heuristic ways to select cumulants a priori running the experiment, e.g. Procgen seems to work better with reward-based similarities, while DCS works better with action-based similarities. See our detailed response to R1 bullet point 1.

---

> > ### Comment · Reviewer_EVg5 · 2022-08-05
> > **Response**
> >
> > After reading the rebuttal, I decided to raise my score to 6.

---

### Official Review · Reviewer_XVwk · 2022-07-17

**Rating:** 5
**Confidence:** 4
**Soundness:** 3 good
**Presentation:** 3 good
**Contribution:** 2 fair

**Summary:**

This paper proposes a behavior similarity-based representation learning method that improves the generalization performance in an offline RL setting. Two key ideas are presented here:
- (1) the proposed behavior similarity is based on the distance of GVF predictions between two states.
- (2) to get a robust estimation of the distance between two GVF signals, it is formulated as a cumulative distribution function under assumptions, which is further converted to K quantile bins and can be learned by self-supervised learning methods in multi-label classification tasks, as defined in Eq. (6).

Procgen and DM Control suite experiments were conducted to demonstrate the effectiveness of the proposed method.

**Questions:**

- (1) Cumulant specification question as I raised in the Weakness part.
- (2) The performance gain issue in Fig. 8, as I raised in the Weakness part.
- (3) Besides the cumulant function definition, we know that the GVF function is also dependent on a target policy, could the authors explain to me how the target policy is defined?
    - If it is different from the behavior policy that collects samples to compose the offline dataset, can the authors explain to me further how to tackle the off-policy learning issue here when you use the TD loss, as shown in Fig. 2?

- (4) I can see the full loss function contains both TD loss and GSF loss in Fig. 2, but there is no section, nor an algorithm that gives a complete description of how exactly the learning process happens. I can find the TD loss in Algorithm 1, line 6, but I am still wondering what is the complete algorithm solution as you summarized in Fig. 2, "Train Agent using L_TD + lambda * L_GSF". Could the authors help me to understand the complete training process?

**Ethics Review Area:**

["I don’t know"]

**Strengths And Weaknesses:**

Strengths
---------------------------------
- The problem setting of studying generalization using an offline dataset, unlike RAD, CURL, and bi-simulation Metric has significant importance in real-world problems.
- The authors demonstrate sufficient knowledge in generalizable RL literature. I really enjoy reading the introductory part.
- The proposed behavior similarity based on GVF predictions makes sense to me that should demonstrate better generalization performance.
- The conversion of robust GVF distance estimation problem to a multi-label (as K bins) self-supervised learning task, is quite nice.

Weaknesses
------------------------------
- The generality of GVF functions. I am concerned about how to specify the cumulant of the GVF function, as this is the key part that determines the generalization performance (as the authors mentioned in lines 314 ~ 319). After checking the paper, specifying the cumulants seem to still require pre-defined using some kind of human heuristic knowledge. Will the authors explain how to resolve the concern of generality of hand-specified cumulants?
- The performance gain (Fig. 8 in Appendix, page 21) seems not significant. When comparing the proposed method to a vanilla PPO in the Procgen benchmark seems very small, and most tasks do not see a significant performance improvement, except for Caveflyer, Leaper, and plunder. Will the authors help to explain any reasons for that?

---

> ### Author Response · Authors · 2022-08-01
> **Response to reviewer XVwk**
>
> Thank you for your constructive comments and expressing valid concerns. We realize that some figures and formulations in the paper might have been confusing, and provide a point-by-point detailed response below.
>
> * **Generality of GVFs:** You brought up an interesting point and a valid concern. You are correct in that the cumulant function is treated as a “hyperparameter” and has to be selected using domain-specific knowledge. However, we have empirical evidence that some cumulant functions are more suitable for certain domains. For example, reward-based GVFs (i.e. value functions) see the largest performance boost in procedurally-generated environments with discrete actions (Procgen). On the other hand, Distracting Control suite results show that reward-based cumulants underperform compared to action-based cumulants, which is consistent with the results of Agarwal et al. (2021) who argued that deterministic environments with continuous action spaces should learn state representations based on distance between near-optimal policies. The underlying reason of why this split happens hides in the structure of the MPDs: in Procgen (see Fig.1) there exists multiple optimal policies due to the nature of the action space and the procedural task variations; the policies also tend to be high-entropy (see Fig.5), which makes similarities between two near-optimal policies quite small. In continuous control, even deterministic methods such as DDPG can find optimal policies, making action-based similarities more suitable in that setting.
> * **Performance of GSF in Fig.8:** You’re right, in that the performance gain versus PPO is less significant in online Procgen than in offline Procgen. However, we want to point out that: 1) PPO is a strong baseline on online Procgen, consistently achieving high scores (Mohanty et al. 2021) and 2) GSFs are proposed in the offline setting, which is where they see most gains (offline Procgen). We included online results in order to be fully transparent and show that PPO+PSEs (action-based similarity) underperforms in online Procgen as well, as opposed to PPO+GSFs with reward-based similarity, which confirms our results from the offline setting. We also believe that since PPO is an on-policy algorithm, GSFs do not fully benefit from vast amounts of data as opposed to using an off-policy method e.g. Soft Actor-Critic. We chose PPO instead of SAC as a baseline as it is the most common baseline on online Procgen and performs better than SAC (see Mohanty et al. 2021 Table 1).
>
> * **Dependence of GVF on policy:** The GVFs are learned similarly to Agarwal et al. (2021): we first split the large dataset on a per-task basis, and then perform one-step TD updates for off-policy evaluation while replacing the reward signal with the chosen cumulant function (see Alg.2). The resulting GVF is therefore computed with respect to the logging policy which, in the case of offline Procgen, is a mixture policy between random uniform and PPO policy after 25M frames.
>
> * **Complete algorithm for using GSF for control:** Apologies for not making the algorithm more clearly visible - it is currently located on page 17 (Alg. 2) and referenced on line 245 of the main manuscript. The idea is that, we first execute Alg.1 to compute $m$ GVFs for $K$ training tasks. We then use these m GVFs to compute the quantile function and label every single data point in the dataset. Finally, we then update the CQL agent together with $\ell_\text{GSF}$ which uses these hypothesized labels in the cross-entropy loss.

---

> > ### Comment · Reviewer_XVwk · 2022-08-08
> > **Will not stick on the generality of GVFs, therefore raise the score.**
> >
> > Hi, excellent response!
> >
> > The toughest question that I raised above may be the generality of GVFs cumulant function design. But one paper can not solve all the problems.  And the proposed generalization performance based on GVF similarity makes sense to me. I will raise my score a little bit.
> >
> > Thanks.

---

### Author Response · Authors · 2022-08-01
**General response**

We thank all reviewers for their constructive comments and hope we addressed all of them in our response below. We have also uploaded a revised version of the manuscript which fixes typos pointed out by reviewers, and adds the UMAP visualization plots requested by reviewer *dP54*.

---

### Meta-Review · Area_Chair_3iXM · 2022-08-25

**Recommendation:** Accept
**Confidence:** Certain

**Metareview:**

This paper studies an interesting problem, and overall the reviewers agreed the exposition and validation are sufficient. We encourage the authors to consider the issues raised by the reviewers and further improve the work in the final version.

**Award:**

No

---

### Decision · Program_Chairs · 2022-09-14

Accept